# Transcriptome Analysis of the *CML* Gene Family in *Bletilla striata* and Regulation of Militarine Synthesis Under Sodium Acetate and Salicylic Acid Treatments

**DOI:** 10.3390/plants14071052

**Published:** 2025-03-28

**Authors:** Kunqian Li, Mengwei Xu, Qingqing Li, Hongwei Li, Ya Xu, Delin Xu

**Affiliations:** Department of Medical Instrumental Analysis, Zunyi Medical University, Zunyi 563099, China; mfkunqian@163.com (K.L.); biologyxmw@126.com (M.X.); 18323158371@163.com (H.L.);

**Keywords:** *Bletilla striata*, *CML* gene family, secondary metabolite, militarine

## Abstract

Calmodulin-like proteins (CMLs) are essential for calcium signal transduction in plants, influencing growth, development, stress responses, and the regulation of medicinal secondary metabolites. Despite their importance, the roles of *CML* genes in *B. striata* have not been characterized. This study aimed to elucidate the composition and function of the *BsCML* gene family in *B. striata*, identifying 38 genes across eight subfamilies. Evolutionary analysis showed that *BsCML* genes are stable and conserved, while functional predictions indicated involvement in environmental stress response, hormone regulation, and circadian rhythms. Expression profiling revealed that *BsCML27* and *BsCML16* were highly expressed during callus culture, suggesting their involvement in growth and development. Notably, *BsCML32* and *BsCML37* exhibited bidirectional regulation of militarine synthesis under sodium acetate (NaAc) and salicylic acid (SA) treatments, with tissue-specific expression strongly correlated (*p* < 0.01) with metabolite accumulation. These findings highlight the significant roles of *BsCML* genes in stress response and secondary metabolite synthesis, providing a foundation for enhancing the medicinal quality of *B. striata*.

## 1. Introduction

The Ca^2+^ signaling system plays a crucial role in regulating plant growth and development, ensuring stability under biotic and abiotic stress conditions [1]. Calcium signals are characterized by the frequency, amplitude, and distribution of calcium concentration fluctuations, which exhibit both spatial and temporal specificity. In response to various environmental stressors or internal variations, plants typically modulate their biological activities by integrating the spatiotemporal dynamics of calcium ion changes [2,3,4]. For instance, the opening and closing of plant stomata, as well as the formation of root hair symbiosis under high-temperature conditions, are regulated by calcium signals [5]. Calcium signaling plays a central role in various aspects of biological regulation. The precise transmission of these extensive signals relies on the involvement of multiple calcium-sensing proteins, including calmodulin (CaMs), calmodulin-like proteins (CMLs), calcineurin B-like proteins (CBLs), and calcium-dependent protein kinases (CDPKs) [6]. The fluctuations of calcium signals are decoded by these calcium sensors, which subsequently interact with downstream transcription factors (TFs), kinases, ion channels, and other components to complete the process of biological signal transduction [7]. The EF-hand motif—a helix-loop-helix structure critical for Ca^2+^ binding—is the defining feature of calcium-sensing proteins [8]. Upon binding to Ca^2+^, this structure undergoes a conformational change, enabling its signal-mediated functions [9,10,11]. Based on these characteristic structural motifs, researchers have identified CML family members in various plant species, thus establishing a foundation for exploring the functional roles of CMLs.

Calmodulin-like proteins (CMLs) play a crucial role in plant growth and development, stress resistance, and the regulation of substance metabolism. For instance, *CML38* interacts with PEP1 receptor 2 to negatively regulate root growth in *Arabidopsis thaliana*, thereby maintaining nitrogen resource balance [12]. Additionally, *CML38* has been identified as a key regulator of root response to hypoxia stress [13]. In addition, *AtCML9* has been shown to promote innate immunity in *A. thaliana* [14]. *MsCML10* interacts with glutathione S-transferase (MsGSTU8) and fructose 1,6-bisphosphate aldolase (MsFBA6) to positively regulate cold tolerance in *Medicago truncatula* [15]. The role of *CMLs* in enhancing plant stress resistance has been widely documented. Overexpression of *MdCML3* increases the resistance of *Malus domestica* callus to high salinity and abscisic acid [16]. Similarly, overexpression of *SlCML37* enhances cold tolerance in *Solanum lycopersicum* [17]. Furthermore, *AhCML69* in *Arachis hypogaea* positively regulates plant disease resistance [18]. To adapt to stress factors or changes in the growth environment, plants often activate specific adaptation mechanisms. As a key medium for plant–environment interactions, the generation and metabolic regulation of secondary metabolites play a critical role in enabling plants to cope with stress. In *A. thaliana*, under oxidative stress, the synthesis of AsA is positively regulated by *AtCML10* to mitigate such stress [19]. For instance, in *Gossypium spp.*, *GbCML45* and *GbCML50* positively regulate lignin accumulation to enhance resistance to Verticillium wilt [20]. These studies suggest that the *CML* family may represent one of the key hub genes involved in the synthesis of secondary metabolites. Furthermore, in studies addressing abiotic stress, it was observed that hormones such as salicylic acid, jasmonic acid (JA), methyl jasmonic acid (MeJA), and abscisic acid (ABA) play significant regulatory roles in the expression of *CMLs*. This finding indicates that *CMLs* may regulate the synthesis of secondary metabolites by responding to these hormonal signals.

To investigate the synthesis mechanism of militarine in *B. striata*, salicylic acid (SA) and sodium acetate (NaAc) were used to treat the callus of *B. striata* in previous studies. Transcriptome and metabolome analyses revealed that the expression of *CML* was significantly altered at certain time points. A comprehensive analysis indicated that *CML* is one of the key genes involved in regulating the synthesis of militarine [21]. Militarine is a glycosidic active compound found in *B. striata*, known for its anti-tumor, anti-inflammatory, hemostatic, whitening, and antioxidant effects [22]. It is listed as the sole medicinal component of *B. striata* in the 2020 edition of the ‘Chinese Pharmacopoeia’ [23], underscoring its significant research value. Despite the pharmacological importance of militarine and the established roles of *CMLs* in other plants, the *CML* gene family in *B. striata* (*BsCML*) remains uncharacterized, and its regulatory role in militarine biosynthesis is unknown. Additionally, the regulatory effects of SA and NaAc on *BsCML* remain poorly understood. In this study, we identified 38 *CML* members from *B. striata* and systematically analyzed their encoded proteins, including physicochemical properties, gene structures, evolutionary relationships, and simple sequence repeat (SSR) loci. Also investigated were the temporal expression patterns of *BsCML* members under SA and NaAc treatments and their regulatory functions in militarine biosynthesis. These results provide valuable insights for further functional characterization of the *BsCML* family and establish an experimental foundation for elucidating the genetic mechanisms underlying militarine biosynthesis.

## 2. Results

### 2.1. Functional Annotation of BsCML

A total of 9996/18,309 differentially expressed genes (DEGs) and 1561/2107 differential metabolites (DEs) were identified from the suspension cells of *B. striata*, including 42 *BsCML*. KEGG and GO annotation analyses indicated that BsCML plays a significant role in calcium ion binding, the MAPK signaling pathway (mitogen-activated protein kinase), and plant–pathogen interactions. These findings suggest that BsCML is involved in plant signal transduction, immune defense, and stress response, thereby maintaining normal physiological functions. Considering the effects of NaAc and SA on growth stress and induction in *B. striata* callus, *BsCML* was selected as a candidate gene for further investigation into the mechanism of secondary metabolite accumulation in *B. striata*.

### 2.2. Identification and Proteins Physicochemical Property Analysis of BsCML

A total of 42 *CML* gene sequences were initially obtained from the transcriptome data of *B. striata*. The CML sequences from three species and the HMM profile were used for alignment and screening. Ultimately, 38 unigenes related to *CML* were identified and designated as *BsCML1* to *BsCML38*. In terms of the physicochemical properties of the BsCML family proteins (Table 1), the ORF length ranged from 71 to 271 amino acids (aa), and the molecular weight varied from 7983.08 to 29,322.21 Daltons (Da). Among these, BsCML8 had the smallest ORF, while BsCML17 had the largest. The theoretical isoelectric point (pI) of the BsCML family ranged from 3.71 to 6.31, with BsCML20 having the lowest and BsCML17 having the highest pI, suggesting that BsCML family proteins are generally acidic. The instability index of most BsCML proteins was greater than 40, while the aliphatic index was less than 100, and the Grand Average of Hydropathy (GRAVY) value was negative, indicating that the BsCML family proteins are generally unstable and hydrophilic. Cellular localization predictions indicated that the family members were primarily distributed in the nucleus and cytoplasm (Table 1). These findings suggest that BsCML proteins may play roles in regulating physiological processes, such as energy metabolism, growth and development, and light responses. Furthermore, signal peptide prediction identified that only BsCML31 contained a signal peptide. In the transmembrane structure prediction, BsCML10 and BsCML31 were found to possess transmembrane regions, suggesting that BsCML31 may be involved in the regulation of light response processes.

### 2.3. Amino Acid Conserved Motifs and Gene Structure Analysis

Analysis of the protein structure of BsCML revealed that the conserved motif structures were similar among members of closely related gene families in the evolutionary tree (Figure 1A). It was found that each BsCML member contained between two and four conserved motifs (Figure 1B). A total of nine motifs were identified through the analysis of conserved motif sequences using MEME (Figure 1D). Among these, the conserved EF-hand domain characteristic sequence (D-x-D-x-D) was found in motifs 1, 2, 3, and 5, and was present in all family members (BsCML1 to BsCML38), further indicating that the BsCML family shared characteristics typical of calcium-sensing proteins. In subsequent Conserved Domain (CD) predictions, three EF-hand conserved domains were identified (Figure 1C), which confirmed these findings.

### 2.4. Phylogenetic Analysis of the BsCML Family

In the phylogenetic analysis of BsCML, the BsCML family was divided into eight subgroups (I-VIII) by comparing the conserved domain characteristics of CML sequences across four plant species using TBtools v1.132 (Figure 2). Within subgroup VI, BsCML10, BsCML33, BsCML16, and BsCML31 displayed a close genetic relationship, which is reflected in their structural similarity, as shown in Figure 1B. Additionally, BsCML members are dispersed among various subfamilies, suggesting that BsCML has maintained relatively complete genetic information throughout evolution.

### 2.5. Detection of EST-SSR

Seven sequences containing SSR loci were identified in 38 *BsCML* members, of which three sequences were dinucleotide repeats and four sequences were trinucleotide repeats, with repeat numbers ranging from a minimum of 3 to a maximum of 11 (Table 2). The SSR frequency (i.e., the ratio of the number of unigenes containing SSRs to the total number of unigenes) of *BsCML* in *B. striata* was 18.42%. The results of PAGE detection of SSR molecular markers (Figure 3) showed that *BsCML* could stably amplify bands, and it was observed that *BsCML11*, *BsCML15*, and *BsCML23* exhibited different banding patterns among different strains of *B. striata*. Additionally, multiple allele bands were isolated for *BsCML15* and *BsCML23.* These results indicate that *BsCML* in *B. striata* exhibits polymorphism at SSR loci, potentially playing an important role in the identification of different *B. striata* germplasm resources.

### 2.6. Cis-Acting Elements of the BsCML Gene

The CML family is a key signaling protein mediated by Ca^2+^, playing an essential role in plant responses and defense against various stress factors. In this study, the upstream 2000 bp promoter region of 38 *BsCML* family members was analyzed, revealing a variety of cis-regulatory elements, including elements responsive to physical factors such as low temperature, drought, hypoxia, and light, as well as hormone-related regulatory elements responsive to SA, auxin, and gibberellin. Additionally, elements involved in the regulation of circadian rhythm were identified (Figure 4). The distribution of these elements suggests that the *BsCML* family may have complex and diverse roles in plant responses to environmental stress, hormone regulation, and circadian clock regulation. Through the presence of these regulatory elements in their promoter regions, *BsCML* genes likely play a significant role in influencing plant physiological responses, metabolic regulation, and enhancing adaptability to environmental changes.

### 2.7. Interactions Among BsCML Proteins

During plant growth and metabolism, regulation often involves multiple genes. The regulatory proteins or biocatalytic enzymes encoded by these genes play synergistic or antagonistic roles to ensure normal organismal development. The function and interactions of BsCML proteins were preliminarily examined by comparison with the *CML* genome proteins of the model plant *A. thaliana*. From the BsCML protein–protein interaction (PPI) network (Figure 5A), it was observed that BsCML32, BsCML24, BsCML23, BsCML28, BsCML13, BsCML37, BsCML26, BsCML19, BsCML18, and BsCML30 exhibit close relationships, with BsCML32 at the core. These results suggest a potential close interaction among *BsCML* genes.

The structure serves as the foundation for functional expression. By analyzing the similarity between the protein sequences of *A. thaliana* and *B. striata* (Figure 5B), insights can be gained to further explore the functions encoded by *BsCML* genes. The results indicated that among these interacting proteins, the corresponding protein structures of *A. thaliana* and *B. striata* displayed high similarity and conservation, suggesting that their functions are also similar.

### 2.8. Expression Patterns of BsCML Under NaAc and SA Treatment and Their Effects on Growth and Secondary Metabolite Accumulation in B. striat

To explore the expression differences in *BsCML* family genes at different growth stages and under elicitor treatments, the expression levels of *BsCML* family genes in *B. striata* were analyzed using transcriptomic FPKM values to generate a heatmap. The results indicated that the expression of the *BsCML* gene family was generally silenced or exhibited a low response to SA and NaAc treatments (Figure 6A). Among these genes, the expression levels of *BsCML1-5*, *BsCML7-8*, *BsCML10-15*, *BsCML19-21*, *BsCML25-26*, *BsCML30*, and *BsCML32-36* were relatively low (FPKM < 150), whereas *BsCML27* showed high expression at various developmental stages. It is speculated that *BsCML27* is involved in the growth and development of *B. striata* callus. These findings suggest that the *BsCML* family is in a low-response state during *B. striata* callus culture and that specific *BsCML* members may play roles in the growth and development of *B. striata* cell suspensions. To verify the reliability of the transcriptome data, six genes were randomly selected for validation, with the *β-actin* gene used as a reference control (Figure 6B). The results showed that, among the six randomly selected genes, the expression trends of five genes were largely consistent between FPKM and qPCR data. However, an inconsistency was observed in the expression trend of *BsCML27*. For example, at 18 days, the FPKM expression pattern of *BsCML27* was SA > control > NaAc, whereas the qPCR pattern was control > SA > NaAc. Overall, the qPCR validation results confirm the reliability of the transcriptomic FPKM data.

In this study, NaAc and SA were used as inducers during the suspension culture of *B. striata* callus. The results indicated that NaAc and SA, as exogenous stress factors, could significantly slow the growth of *B. striata* callus to varying extents. Notably, the inhibitory effect of NaAc was highly significant at 42 and 45 dpi (*p* < 0.005) (Figure 7A). Moreover, the accumulation of metabolites in the callus treated with NaAc and SA was notably altered. It is worth mentioning that the elicitors significantly inhibited (*p* < 0.01) militarine accumulation at 27 and 33 dpi, whereas SA treatment significantly (*p* < 0.01) promoted militarine accumulation at 45 dpi (Figure 7B). For the accumulation of Dactylorhin A, the inhibitory effects of NaAc and SA were also significant (*p* < 0.01), particularly at 30 and 42 dpi (Figure 7C). This alteration in secondary metabolite levels reflects the regulation of callus response to stress factors. Notably, the significant effects of the elicitors on the accumulation of militarine and Dactylorhin A were observed after 12 dpi, indicating the time required for the callus to respond to stress factors. These results demonstrate that NaAc and SA can significantly influence the growth and metabolism of *B. striata*, exhibiting varying degrees of promotion and inhibition on the synthesis of secondary metabolites.

### 2.9. Analysis of BsCML Regulation on Militarine and Dactylorhin A Synthesis Under Elicitor Treatment

Based on the metabolic relationships of militarine [22] (Figure 8A), this study analyzed militarine and its derivative, dactylorhin A. A significant inverse accumulation trend between the two compounds was observed during the 0–45 dpi cultivation period (Figure 8B). Correlation heatmap analysis of *BsCML* expression and militarine accumulation in *B. striata* callus at four time points (Figure 8C) revealed distinct expression patterns under different treatments. In the CK group, 21 *BsCML* genes negatively regulated militarine biosynthesis, with *BsCML32* (r = −0.992) and *BsCML37* (r = −0.993) showing significant negative correlations (*p* < 0.01). In the NaAc treatment group, 28 genes were positively correlated with militarine biosynthesis, with *BsCML31* (r = 0.955) showing a significant correlation (*p* < 0.05). In the SA treatment group, 29 genes positively regulated militarine biosynthesis, with *BsCML11* (r = 0.984), *BsCML21* (r = 0.985), and *BsCML37* (r = 0.958) demonstrating significant effects at *p* < 0.05. In the dactylorhin A biosynthesis correlation analysis (Figure 8D), *BsCML* genes in the CK, NaAc, and SA groups mainly exhibited positive regulatory effects. In the SA group, *BsCML17* (r = 0.985), *BsCML29* (r = 0.976), and *BsCML34* (r = 0.971) significantly promoted dactylorhin A biosynthesis (*p* < 0.05). Notably, *BsCML11* strongly regulated both militarine and dactylorhin A biosynthesis.

These results identify *BsCML11*, *BsCML32*, and *BsCML37* as key regulators of militarine synthesis, with bidirectional responses to SA and NaAc treatments. These genes represent prime candidates for metabolic engineering to enhance *B. striata*’s medicinal value. Overall, the regulatory patterns of *BsCML* on militarine and its derivatives lacked consistency, likely due to differences in accumulation patterns and indirect metabolic relationships. These findings suggest that *BsCML* family members play a key role in regulating militarine biosynthesis, which can be either inhibited or promoted under SA and NaAc treatments. This phenomenon may be linked to the activation or suppression of specific transcription factors. Moreover, cis-acting element prediction (Figure 4) identified SA-responsive elements, suggesting that the shift from negative regulation (CK) to positive regulation (SA and NaAc) in most genes could be mediated by these elements.

### 2.10. Analysis of BsCML Expression and Militarine Accumulation in Different B. striata Tissues

To investigate the regulatory roles of *BsCML11*, *BsCML32*, and *BsCML37* in militarine biosynthesis, their expression patterns were analyzed in the roots, tubers, flowers, and leaves of three *B. striata* varieties with distinct phenotypes (Figure 9A). The results revealed significant tissue-specific expression patterns for all three genes (Figure 9B). Notably, the expression levels of *BsCML11* and *BsCML37* in tubers followed the order type3 > type2 > type1, with significantly higher expression in the tubers of purple-flowered varieties (type2 and type3) compared to other tissues (*p* < 0.01 or *p* < 0.0001). Since tubers are the primary medicinal part of *B. striata*, the high expression of *BsCML11* and *BsCML37* may contribute to the superior medicinal value of purple-flowered varieties over yellow-flowered ones. Additionally, *BsCML11* and *BsCML37* exhibited similar expression trends and levels across the three *B. striata* varieties, suggesting that their functions may be conserved or involved in essential plant metabolic processes. For example, *BsCML11* and *BsCML37* showed extremely low expression in type2 flowers but significantly higher expression in the tubers of type2 and type3 compared to type1, which may correlate with differences in flower color and medicinal efficacy. In contrast, the expression of *BsCML32* in the tubers and roots of type1 was significantly lower than in type3, implying its potential role in the differential metabolite accumulation between these varieties. These findings underscore the diversity of *BsCML* expression patterns and their tissue-specific variations, suggesting their regulatory roles in the accumulation and metabolic processes of *B. striata*.

Militarine (tuber content > 2.0%) is the primary marker for assessing the medicinal value of *B. striata*. LC-MS analysis showed that militarine accumulates in roots, tubers, flowers, and leaves, with the highest concentrations found in rootlets and tubers (Figure 9C). The militarine content in the tubers of all three *B. striata* varieties exceeded 20 mg/g, suggesting that, in addition to purple-flowered varieties, certain yellow-flowered varieties also have medicinal value.

Integrated analysis of gene expression (Figure 9B) and militarine content (Figure 9C) provides evidence for the regulatory role of *BsCML* in militarine biosynthesis. For instance, the militarine content in the flowers of all three *B. striata* varieties was extremely low, with nearly undetectable levels in type2 flowers, consistent with the expression patterns of *BsCML11* and *BsCML37*. Additionally, the higher expression of *BsCML32* in Type3 correlates with its greater militarine accumulation compared to type1, further supporting its regulatory function. In type1 and type2 *B. striata*, the expression of *BsCML32* exhibited an inverse relationship with militarine accumulation. Pearson correlation analysis revealed a negative correlation between the two (r = −0.418, *n* = 12), further confirming that *BsCML32* negatively regulates militarine biosynthesis at the *p* < 0.01 level. Furthermore, NaAc treatment induced *BsCML* expression in callus tissues (Figure 8B, CK group). These findings indicate that *BsCML* is involved in the regulation of militarine biosynthesis, affecting its tissue-specific accumulation, and that this process is influenced by external factors.

## 3. Discussion

### 3.1. Evolutionary Analysis of the BsCML Family

In this study, 38 BsCML family members with EF-hand motifs were identified in *B. striata* (Table 1). Changes in the number of gene family members are often considered important indicators of gene evolution. Among monocotyledonous plants, 32 *CML* genes were identified in *Oryza sativa* [24], 39 in *Saccharum officinarum* [25], 54 in *P. equestris* and *D. officinale* [26], and 80 in *Hordeum vulgare* [27]. For dicotyledonous plants, 44 *CML* genes were identified in *Cucumis sativus* [28], 52 in *S. lycopersicum* [29], and 58 in *M. domestica* [16]. These studies indicate that the number of *BsCML* members has not undergone a significant reduction or expansion. Furthermore, the structural characteristics of the proteins encoded by *BsCML* are consistent with the evolutionary patterns observed in various plant lineages. For instance, the ring structure represented by the D-x-D-G-D-G sequence in motif 1 (Figure 1D) is consistent with the EF-hand domains found in CMLs of monocots, bryophytes, and early-diverging eudicots. Similarly, the D-x-D-G-D structure in motif 2 (Figure 1D) aligns with the EF-hand domains present in early-diverging angiosperms and superasterids [30]. During the evolution of *B. striata*, the evolutionary trends of the eight BsCML subfamilies (Figure 2) show similarities to the phylogenetic tree constructed for 41 plant species [31]. These results suggest that the *BsCML* gene family has remained relatively stable throughout evolution.

### 3.2. Expression Pattern and Functional Analysis of the BsCML Gene Family

As a widely cultivated perennial medicinal orchid, the growth and development of *B. striata*, as well as the accumulation of its medicinal components, are influenced by various factors, including temperature, light quality [32], endophytic fungi [33], and soil microorganisms [34]. The diversity of these external environmental factors is a key contributor to the differential expression of *CML* genes in different tissues and organs of plants. In this study, most members of the *BsCML* gene family in *B. striata* callus under suspension induction exhibited low expression levels (Figure 6A). It is speculated that, under conditions characterized by a short cultivation period and mild environmental factors, the activation signals for the *BsCML* gene family were limited, resulting in their overall low expression. It is noteworthy that *BsCML27* (Figure 6A) is highly expressed at all developmental stages, suggesting its involvement in the growth and development of *B. striata* callus. Several studies have indicated that *CMLs* play significant roles in regulating plant growth and development. For instance, *Arabidopsis CML39* is involved in seed development, germination, and fruit development [35]. Additionally, *CpCML15* interacts with *PP2C46* and *PP2C65* to promote the fruit ripening of *Pseudocydonia sinensis* [36]. Moreover, *MtCML42* promotes the flowering of *M. truncatula* by regulating the expression of *ABI5* (Abscisic Acid Insensitive 5) and *FT* (Flowering Locus T) genes [37]. These findings underscore the crucial role of *CMLs* in plant growth and suggest that their function is modulated through interactions with multiple genes. In plants, the expression of *CMLs* exhibits tissue- and organ-specific patterns, as demonstrated by the differential expression observed in *Passiflora edulis* [38] and *M. truncatula* [39]. This evidence further suggests that the spatial expression patterns of *CML* genes are closely linked to the synthesis of secondary metabolites. In this study, the expression levels of *BsCML11* and *BsCML37* in the tubers of two *B. striata* specimens were significantly higher than those in other tissues (Figure 9B). This suggests that *BsCML11* and *BsCML37* may be involved in the accumulation of secondary metabolites in tubers. Previous studies have shown that overexpression of *VaCML65* in wild grape (*V. amurensis*) cell cultures can increase the content of stilbenes [40], while overexpression of *RrCML13* in the roots of *R. roxburghii* enhances the accumulation of AsA [41]. These studies highlight the significance of *CML* expression in the biosynthesis of secondary metabolites.

It is well established that gene families of the same type in different plant species often share similar structures, which typically results in functional similarities. This characteristic provides a useful approach for studying the functions of newly identified gene families. In this study, the BsCML protein interaction network analysis revealed close interactions among family members (Figure 5A), and structural similarities between BsCML proteins and the corresponding *Arabidopsis* CML members were also observed (Figure 5B). Previous studies have demonstrated that AtCML13 and AtCML14 interact with proteins containing isoleucine–glutamine domains in *A. thaliana* [42]. These interactions are essential for regulatory roles in growth and development, contributing to cytoskeletal remodeling and intracellular transport [43]. Through structural similarity matching, AtCML13 was found to correspond to BsCML27 in their interaction networks, providing strong evidence that BsCML27 may play a role in regulating the growth and development of *B. striata*. Additionally, *AtCML37* and *AtCML42* exhibit antagonistic interactions in response to drought and salt stress, modulating the plant’s stress response via the regulation of gibberellin and auxin signaling pathways [44]. By examining the relationships where AtCML37 corresponds to BsCML13 and AtCML42 corresponds to BsCML24 (Figure 5A), these findings further illustrate the potential interactions among BsCML members and support their involvement in plant stress responses.

### 3.3. Prospective Applications of Inducer-Mediated CML Expression in Regulating Secondary Metabolite Biosynthesis

Secondary metabolites are a class of small molecules produced by plants to adapt to environmental stresses, typically serving as mediators of plant–environment interactions [45,46,47]. The involvement of *CML* genes in responses to various environmental stress factors suggests their critical role in the biosynthesis of secondary metabolites. For example, overexpression of *VaCML92* in *Vitis amurensis* improves cold stress tolerance and enhances stilbene synthesis by 7.8- to 8.7-fold [48]. Similarly, exogenous calcium treatment induces the expression of *CmCML11* and *CmCAMTA5*, resulting in elevated accumulation of γ-aminobutyric acid in melon [49]. These studies highlight the importance of exploring plant *CML* functions through the application of inducers. In the current study, treatment with NaAc and SA altered the accumulation of militarine and dactylorhin A in callus cultures at specific time points (Figure 7B,C), reflecting the responsiveness of *BsCML* to hormones and salt stress, and suggesting its potential regulatory role in militarine biosynthesis under stress conditions. Several studies have demonstrated the role of *CMLs* in regulating glycoside biosynthesis. For instance, overexpression of *CML42* in *A.thaliana* enhances the accumulation of glucosinolates [50], whereas *CML42* also exerts a negative regulatory effect on the synthesis of specific glycosides.

SA is a crucial chemical signal in plants, playing an important regulatory role. Studies have shown that SA can enhance plant tolerance to salt stress, drought, and heavy metal stress by activating the antioxidant enzyme system, regulating stomatal movement, stabilizing cell membranes, and improving photosynthetic efficiency [51,52]. SA plays a crucial role in plant stress regulation and has been widely utilized to induce the synthesis of secondary metabolites. For example, SA induces the synthesis of total phenols and anthocyanins in *V. vinifera* fruits [53], significantly promotes the accumulation of glucosinolates in *Brassica oleracea* [54], and enhances the accumulation of alkaloids in *Hemerocallis citrina* [55] and the marine microalga *Arthrospira platensis* [56]. In this study, it was found that the *BsCML* gene family can respond to SA and potentially influence the synthesis of secondary metabolites. For instance, after the CK group was treated with SA and NaAc (Figure 8B), the role of *BsCML2*, *BsCML9*, *BsCML12*, and *BsCML37* in regulating militarine synthesis shifted from negative to positive regulation. This finding suggests that SA induction plays a crucial role in the regulation of militarine synthesis by *BsCML* genes and highlights the diverse expression patterns of *BsCML* members in the regulation of secondary metabolite synthesis. This result aligns with the presumed presence of SA response elements (Figure 4). Furthermore, environmental factors such as light conditions, plant hormones, and various ions also affect the expression of *CML* genes [11]. In the present study, the sensitivity of *CMLs* to inducer responses was also observed under NaAc salt treatment. These observations indicate that *CML* genes are highly sensitive to external environmental cues and can play a significant role in regulating plant growth and metabolism when subjected to appropriate biotic or abiotic stimuli.

## 4. Materials and Methods

### 4.1. Experimental Materials

The capsules of *B. striata* were collected from the nursery at Zunyi Medical University (27°42′ N, 107°01′ E), located in the Xinpu District of Zunyi City, Guizhou Province, China. The powdered seeds were cultured in suspension for 30 days using Murashige and Skoog (MS) medium supplemented with 1 mg/L 6-benzylaminopurine (6-BA), 2 mg/L 2,4-dichlorophenoxyacetic acid (2,4-D), 0.5 mg/L naphthaleneacetic acid (NAA), and 30 g/L sucrose. The cultures were maintained in sterile bottles (35 mL medium per 200 mL bottle) at a temperature of 25 ± 1 °C under dark conditions, shaking at 120 rpm. After the initial culture period, well-grown calli were selected for suspension subculture at 1 g per bottle. The subculture medium consisted of half-strength MS (1/2MS) supplemented with 1 mg/L 6-BA, 3 mg/L 2,4-D, 0.5 mg/L NAA, and 30 g/L sucrose. Subcultures were maintained under the same conditions (25 ± 1 °C, dark culture, 120 rpm shaking speed), and the medium was refreshed every 15 days [57]. From the second subculture (0 days post-inoculation, 0 dpi) to 45 dpi, samples were collected at random every 3 days, and growth was monitored throughout the period. After the second subculture, 150 μmol/L NaAc and 15 μmol/L SA were added separately, and three samples were randomly selected at 3 dpi, 18 dpi, 21 dpi, and 36 dpi [21]. Total RNA was extracted from each sample for transcriptome sequencing, and the *CML* gene set of *B. striata* was screened from the sequencing results for subsequent analysis.

### 4.2. Detection of Secondary Metabolite Accumulation in Callus by HPLC

Metabolites from suspension-cultured cells were collected at 3, 18, 21, and 36 dpi, with three biological replicates at each time point, and quantified using the Waters e2695 High-Performance Liquid Chromatography (HPLC) system (Waters, Milford, MA, USA). After precise weighing, the standards (militarine and dactylorhin A) were dissolved in methanol to prepare individual high-concentration standard solutions. The concentration of both standard solutions was 0.5 mg/mL, and they were stored according to the manufacturer’s instructions. Separation of the two components was performed on an ACQUITY UPLC BEH C18 column (2.1 × 100 mm, 1.7 µm). The column temperature was maintained at 30 °C, and the injection volume was set to 5.00 µL. The flow rate of the mobile phase was 0.3 mL/min. Mobile phase A consisted of 0.1% formic acid aqueous solution, while mobile phase B was acetonitrile. The gradient elution program was as follows: 0–10 min, 20% A and 80% B; 10–25 min, 50% A and 50% B; 25–27 min, 95% A and 5% B; 27–30 min, 95% A and 5% B; 30–32 min, 20% A and 80% B.

### 4.3. Characteristic Analysis Based on Nucleotide and Amino Acid Sequences

#### 4.3.1. Identification of *BsCML* Family Members in *B. striata*

In order to identify the *CML* gene family members of *B. striata*, the *CML* gene family sequences of *A. thaliana* were obtained from the TAIR database (https://www.arabidopsis.org/), and the *CML* gene family sequences of *Dendrobium catenatum* and *Phalaenopsis equestris* were downloaded from NCBI. The *CML* members of *B. striata* were screened using TBtools v1.132 for conserved domain alignment. To further verify the screening results, the hidden Markov model (PF13499) of *CML* was downloaded from InterPro95.0 (http://www.ebi.ac.uk/interpro/entry/pfam/#table (accessed on 2 August 2023)), and TBtools [58] was used to search all *CML* sequences of *B. striata*.

#### 4.3.2. Analysis of Conserved Motifs and Domains of BsCML

To explore the sequence characteristics of *BsCML* and the evolutionary relationships among its family members, a phylogenetic tree of the *BsCML* gene family was constructed and visualized using TBtools. The MEMESuite5.5.3 tool (https://meme-suite.org/meme/index.html (accessed on 3 August 2023)) was employed to analyze the conserved motifs of CML sequences in *B. striata*, with the number of motifs set to 20. Subsequently, the EF-hand domain of BsCML was identified using the CD-Search function of NCBI and visualized by TBtools.

#### 4.3.3. Analysis of Physical and Chemical Properties of Proteins

To understand the properties of the proteins encoded by *B. striata* CML, ORFfinder (https://www.ncbi.nlm.nih.gov/orffinder/ (accessed on 2 August 2023)) was used to analyze the open reading frames (ORFs) of BsCML. ExPASy (https://web.expasy.org/protparam/ (accessed on 2 August 2023)) was employed to predict molecular weight (MW/Da), amino acid size (AA), isoelectric point (pI), instability index, alpha-helix content, and other physical and chemical properties [59]. The secondary structure of the proteins was predicted using SOPMA (https://npsa.lyon.inserm.fr/cgi-bin/npsa_automat.pl?page=/NPSA/npsa_sopma.html (accessed on 2 August 2023)) [60].

#### 4.3.4. Detection and Validation of EST-SSR

To explore the conservation of *CML* in different strains of *B. striata*, the genes containing SSR loci were analyzed. Genomic DNA was extracted using the CTAB method and diluted to a concentration of 50 ng/μL for EST-SSR detection. The SSR loci of 38 *CML* sequences were detected using the NWISRL online server (https://ssr.nwisrl.ars.usda.gov/ (accessed on 4 August 2023)), with parameters set to default values. The online tool Primer3 Plus (https://www.primer3plus.com/index.html (accessed on 4 August 2023)) was used to design specific primers for SSR loci (Table 2), followed by PCR amplification and PAGE detection of the results. After electrophoresis, the bands were visualized using silver staining and photographed.

### 4.4. Functional Prediction Based on BsCML Sequence Features

#### 4.4.1. Subcellular Localization, Signal Peptide Identification, Transmembrane Structure, and Cis-Element Analysis of the *BsCML* Gene

The subcellular localization of the *CML* gene was predicted using WoLF PSORT (https://wolfpsort.hgc.jp/ (accessed on 22 January 2024)) [61]. The signal peptide of the CML was identified using SignaIP-4.1 (https://services.healthtech.dtu.dk/services/SignalP-4.1/ (accessed on 22 January 2024)). The transmembrane structure was predicted using Deep TMHMM (https://dtu.biolib.com/DeepTMHMM (accessed on 22 January 2024)). To analyze the homeostatic elements within the 2000 bp promoter region upstream of the *BsCMLs* gene, the PlantCARE database (http://bioinformatics.psb.ugent.be/webtools/plantcare/html/ (accessed on 21 January 2024)) was used, and the results were visualized in Tbtools, with each element represented by a 20 bp window for localization purposes.

#### 4.4.2. Analysis of Evolutionary Characteristics of the *BsCML* Gene Family

To analyze the evolutionary relationships within the 38 *CML* gene families of *B. striata*, 102 *CML* amino acid sequences from *A. thaliana*, *Dendrobium nobile*, and *Phalaenopsis* were used as references. Specifically, 29 *A. thaliana*, 40 *D. catenatum*, and 33 *P. equestris CML* sequences were downloaded, along with the identified *BsCML protein* sequences, to construct an interspecies phylogenetic tree. MAGE11 was used to process the data (parameter settings: Neighbor-Joining method, Bootstrap replications set to 1000, with other parameters set to default), followed by visualization using iTOL (https://itol.embl.de/ (accessed on 31 July 2023)). Finally, subfamily classification was performed based on the conserved domain characteristics of these genes.

#### 4.4.3. Interaction Network Analysis of BsCML Family Proteins

To analyze the protein–protein interaction (PPIs) and gene ontology (GO) of the BsCML family proteins, the online tool STRING 12.0 (http://string-db.org/ (accessed on 22 January 2024)) [62] was used with default parameters, using *A. thaliana* as a reference plant. Subsequently, CLUSTALW (https://www.genome.jp/tools-bin/clustalw (accessed on 28 September 2024)) and ESPript 3.0 (https://espript.ibcp.fr/ESPript/ESPript/index.php (accessed on 28 September 2024)) [63] were employed for sequence alignment analysis of potentially homologous proteins.

### 4.5. Functional Verification and Analysis of the BsCML Gene Family

#### 4.5.1. Expression Pattern Analysis and Quantitative Fluorescence PCR Validation of *BsCML*

In a previous study, the content of militarine was measured across multiple growth stages, revealing that four specific time points (3 dpi, 18 dpi, 21 dpi, 36 dpi) served as critical inflection points for militarine synthesis [57]. At these time points, the expression pattern of the *BsCML* gene family was visualized using ChiPlot (https://www.chiplot.online/ (accessed on 28 April 2024)), based on the FPKM values of *CML* transcripts in *B. striata* callus.

To verify the expression of *CML* genes represented by the FPKM values from the transcriptome, the first strand of cDNA was synthesized by reverse transcription using the TIANGEN kit, with 1 µg of total RNA from qualified *B. striata* suspension cells at 18 dpi and 21 dpi as the template. The qPCR reaction was performed on the AGS4800 real-time quantitative PCR instrument. The total volume of the qPCR reaction mixture was 10 µL, consisting of 1 µL cDNA template (with a concentration of 55–60 ng/µL), 5 µL SYBR Green qPCR Master Mix, 0.2 µL of each primer (10 µmol/L), and 3.6 µL ddH_2_O. Primer information is shown in Table 3. Thermal cycling conditions were as follows: initial denaturation at 95 °C for 30 s, followed by 40 cycles of 95 °C for 10 s, 55–60 °C for 30 s, and 72 °C for 30 s (fluorescence acquisition). The final step was at 95 °C for 15 s. Gene expression was normalized to the *β-actin* reference gene (GenBank: XM_045392100).

To explore the regulatory role in the synthesis of militarine and dactylorhin A, Pearson correlation analysis (PPMCC) was performed using SPSS 29.0 between the measured metabolite concentrations (3 biological replicates and 3 technical replicates) and the expression of *BsCML* transcripts. A correlation heatmap was then generated to visualize the data. Through comprehensive analysis, genes significantly associated with the accumulation of secondary metabolites were identified within the *BsCML* family.

#### 4.5.2. Expression Profile Analysis of *BsCML* in Different *B. striata* Tissues

To investigate the regulation of *BsCML* family members in the synthesis of secondary metabolites and their expression patterns in perennial *B. striata* plants, three *BsCML* family members—*BsCML11* *, *BsCML32* **, and *BsCML37* ** (* *p* < 0.05, ** *p* < 0.01 level correlation)—which were significantly associated with militarine synthesis, were selected for further study. RT-PCR was conducted on four tissues: flowers, leaves, tubers, and roots, from three types of *B. striata* plants that had been cultivated for over three years. The 2^−ΔΔCt^ method was used to calculate gene expression relative to *β-actin*.

#### 4.5.3. LC-MS Detection of Secondary Metabolite Accumulation in Different *B. striata* Tissues

The roots, tubers, leaves, and flowers from three different *B. striata* plants were collected, and the metabolites were quantitatively analyzed using an Agilent 6545 Q-TOF liquid chromatography-mass spectrometry (LC-MS) system. The analytes were quantified in both positive and negative ion detection modes using multiple reaction monitoring (MRM). The three standards were accurately weighed and dissolved in methanol to prepare individual high-concentration standard solutions (militarine at 0.25 mg/mL and dactylorhin A at 0.5 mg/mL). The following conditions were used: Agilent Eclipse Plus C18 column (2.1 × 100 mm, 1.8 µm); flow rate: 0.5 mL/min; mobile phase: water with 0.1% formic acid (A) and 100% acetonitrile (B); column temperature: 30 °C; detection wavelengths: 220 nm and 254 nm; injection volume: 5.00 µL. The gradient elution was as follows: 0–0.5 min, 90% A and 10% B; 0.5–6 min, 100% B; 6–8 min, 90% A and 10% B.

By comprehensively analyzing the relationship between gene expression and metabolite accumulation in *BsCML* family members, we identified *BsCML* genes closely involved in militarine synthesis, providing an experimental basis for future studies on militarine biosynthesis.

## 5. Conclusions

In this study, 38 BsCML family members containing EF-hand motifs were identified from *B. striata* based on genomic sequencing and were classified into eight subfamilies (I–VIII). Analysis of *BsCML* gene structures, conserved protein domains, and phylogenetic relationships indicated that the inheritance of *BsCML* genes during evolution is highly conserved. Interaction analysis revealed a close interaction network within the BsCML family. Although most *BsCML* members exhibit low or negligible expression, they show significant tissue-specific differences and can be activated to varying degrees by elicitors. Upon treatment with elicitors such as NaAc and SA, the regulatory roles of *BsCML* in the accumulation of secondary metabolites changed significantly, suggesting that the metabolite accumulation in *B. striata* is regulated through the interaction between *BsCML* genes and the external environment. Furthermore, the expression patterns of *BsCML11*, *BsCML32*, and *BsCML37* exhibited a high degree of similarity to the trends observed in militarine accumulation, suggesting that the differential expression of *BsCML* genes is closely related to the accumulation of metabolites in *B. striata*. These genes may serve as candidate genes for further investigation into the biosynthetic mechanisms of militarine. The results of this study have identified the *BsCML* family in *B. striata* and systematically explored their potential functions, thereby laying a foundation for the future utilization of *BsCML* molecular resources.

## Figures and Tables

**Figure 1 plants-14-01052-f001:**
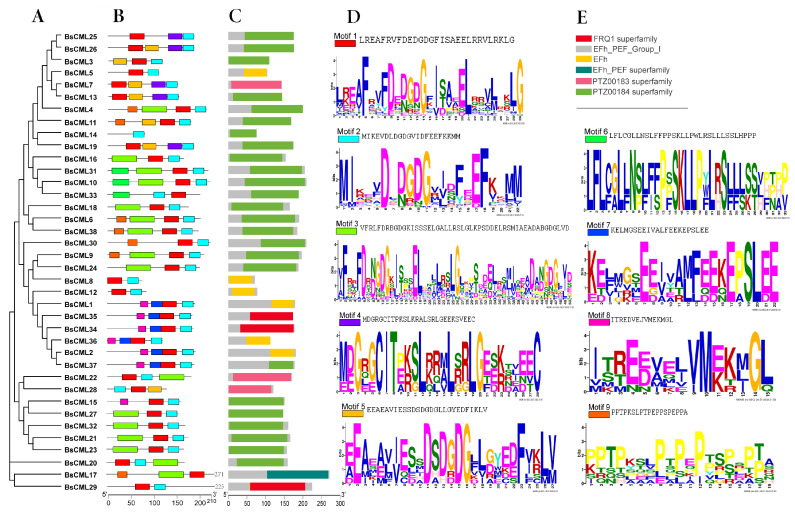
Phylogenetic and structural analysis of BsCML proteins. (**A**) Phylogenetic tree of the BsCML family. (**B**) Motif analysis of BsCML proteins. (**C**) Prediction of conserved domains (CDs) in BsCML proteins. (**D**) Sequence logos corresponding to panel (**B**), where the height of each letter represents the bit score of the respective amino acid, with higher bit scores indicating greater conservation. (**E**) Six predicted structural motifs identified in panel (**C**).

**Figure 2 plants-14-01052-f002:**
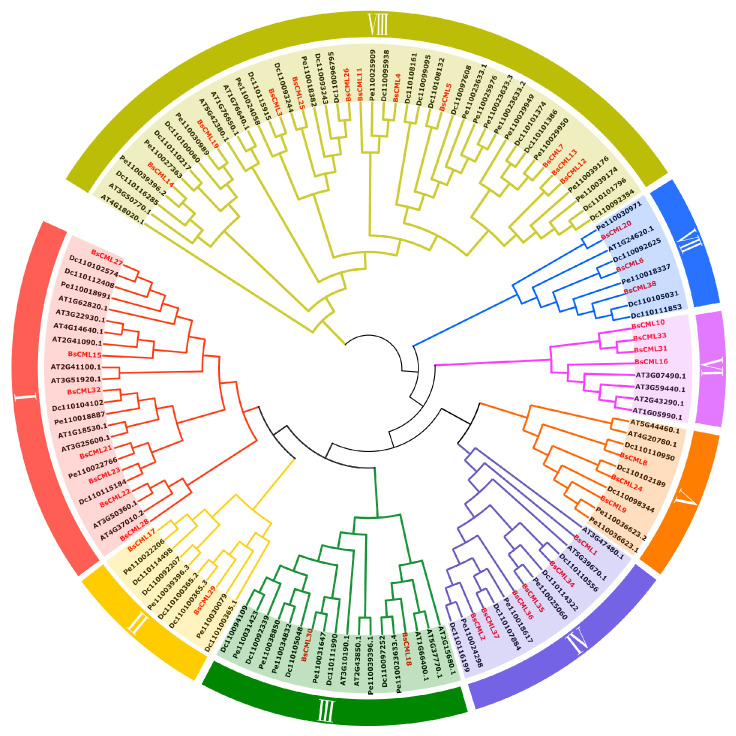
Phylogenetic relationships of BsCML with CML families from *A. thaliana* (At), *D. catenatum* (Dc), and *P. equestris* (Pe). Members of the BsCML family are highlighted in red and distributed across subfamilies I–VIII, with each subfamily denoted by a distinct color.

**Figure 3 plants-14-01052-f003:**
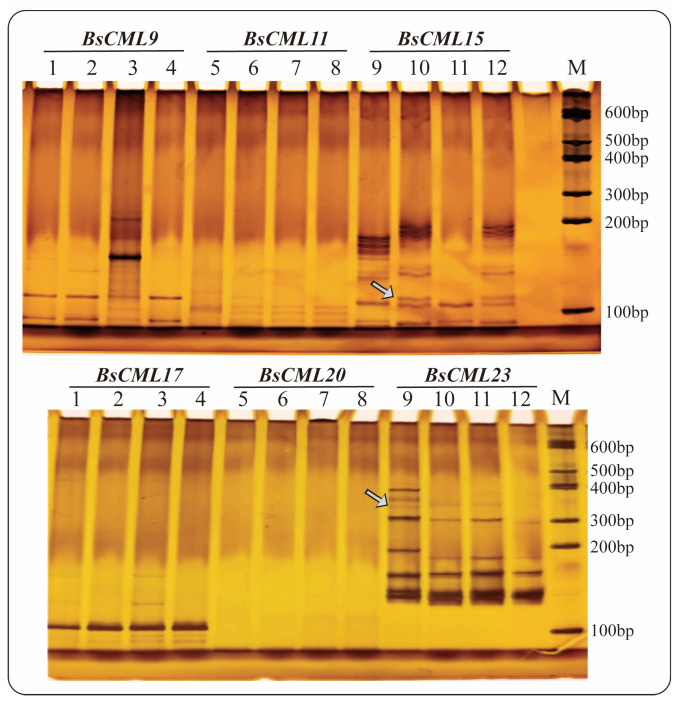
PAGE detection of SSR molecular marker amplification results for *BsCML* genes in four *B. striata* varieties. Lane numbering in the figure corresponds to the following PCR templates: Lanes 1, 5, 9: Genomic DNA from ZMU-Bs001. Lanes 2, 6, 10: Genomic DNA from ZMU-Bs002. Lanes 3, 7, 11: Genomic DNA from ZMU-Bs003. Lanes 4, 8, 12: Genomic DNA from ZMU-Bs004. *BsCML15* and *BsCML23* show polymorphic bands (arrows), indicating SSR locus diversity.

**Figure 4 plants-14-01052-f004:**
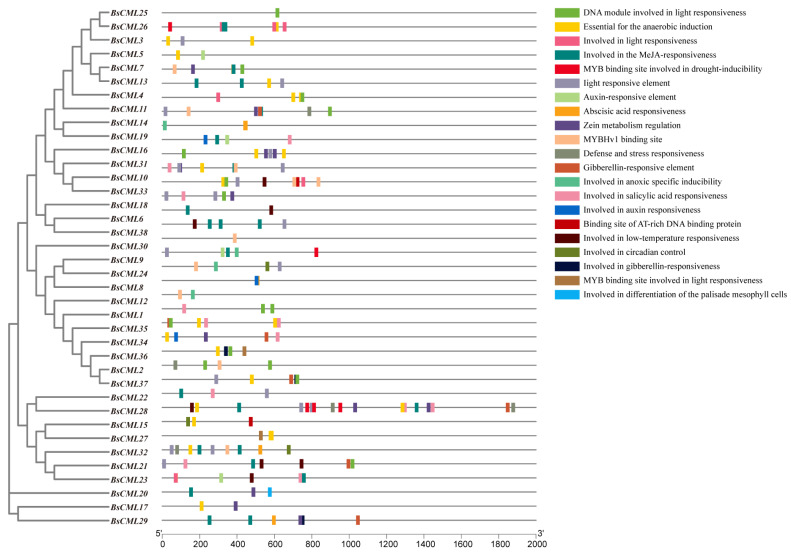
Potential cis-acting elements in the *BsCML* gene.

**Figure 5 plants-14-01052-f005:**
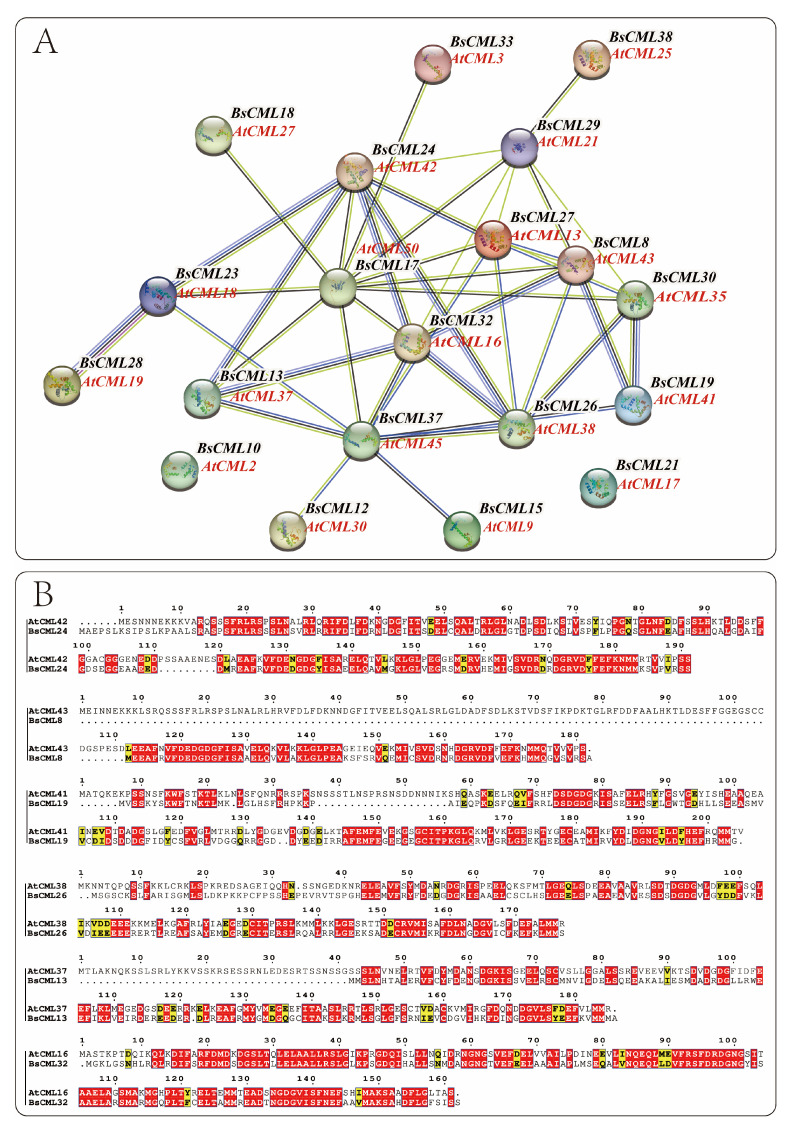
Protein interaction network of BsCML and comparative analysis of sequence similarity with Arabidopsis CML. (**A**) Protein–protein interaction network of BsCML: Using AtCML as the reference genome, spheres (nodes) represent BsCML and AtCML proteins from the model plant Arabidopsis. The thickness of the connecting lines indicates the strength of interactions between two proteins. (**B**) Comparative analysis of protein sequence similarity between *B. striata* and Arabidopsis: Based on the protein interaction relationships shown in panel A, six pairs of protein sequences were analyzed. Red regions indicate highly conserved domains or critical functional sites, while yellow regions represent areas with moderate conservation or weak similarity.

**Figure 6 plants-14-01052-f006:**
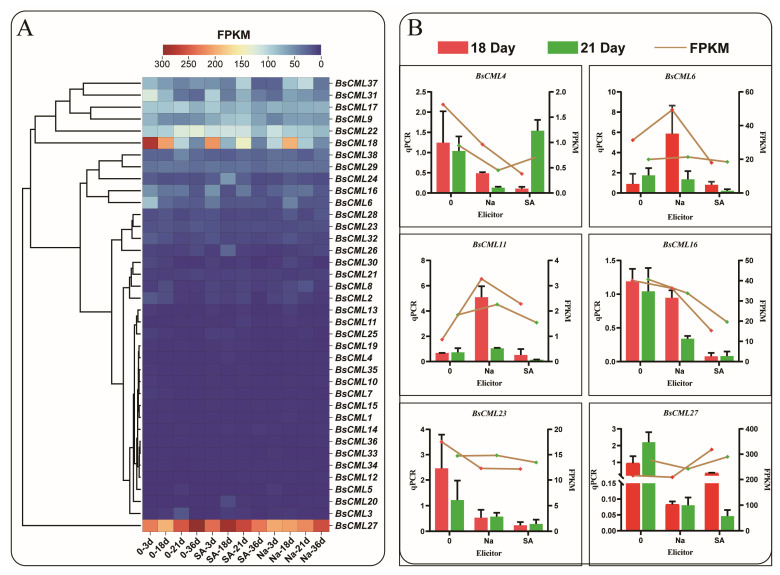
Expression profiles of *BsCML* during suspension culture of callus and qPCR validation. (**A**) Expression patterns of *BsCML* family members at 3 dpi, 18 dpi, 21 dpi, and 36 dpi: ‘Na’ represents treatment with 150 μmol/LNaAc, ‘SA’ denotes treatment with 15 μmol/L SA, and ’0’ indicates the control group without any inducer. (**B**) qPCR validation of the expression levels of six selected *BsCML* genes.

**Figure 7 plants-14-01052-f007:**
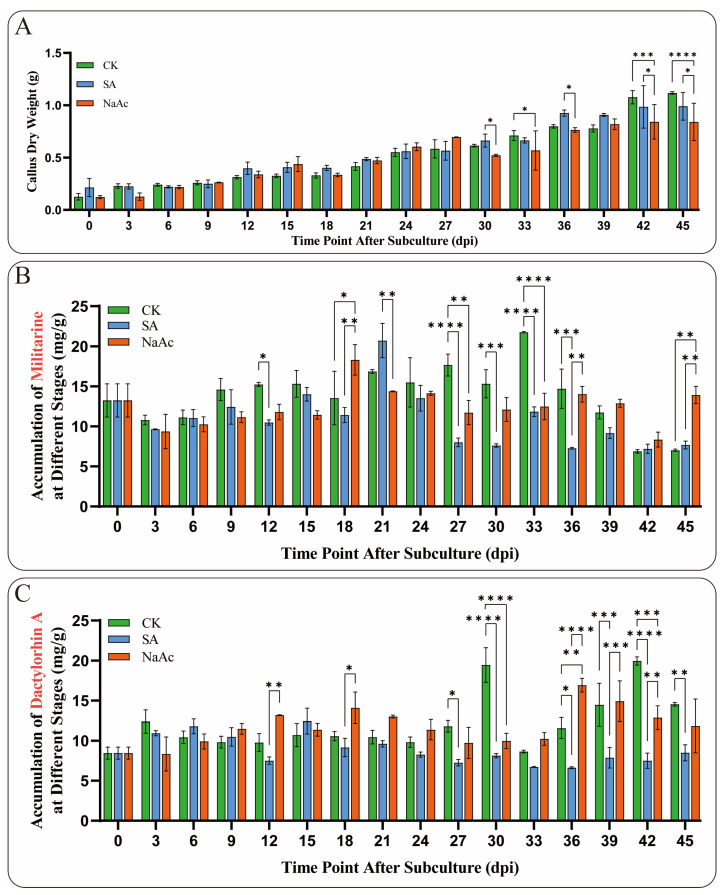
Growth and metabolite accumulation of *B. striata* callus treated with 15 μmol/L SA and 150 μmol/L NaAc. (**A**) Dry weight growth trend of *B. striata* callus cells in the elicitor and control (CK) groups from 0 to 45 dpi. (**B**) Changes in the accumulation of militarine in the induced callus. (**C**) Changes in the accumulation of Dactylorhin A in the callus treated with induction. Asterisks denote significance levels: * *p* < 0.05, ** *p* < 0.01, *** *p* < 0.001, **** *p* < 0.0001 (one-way ANOVA).

**Figure 8 plants-14-01052-f008:**
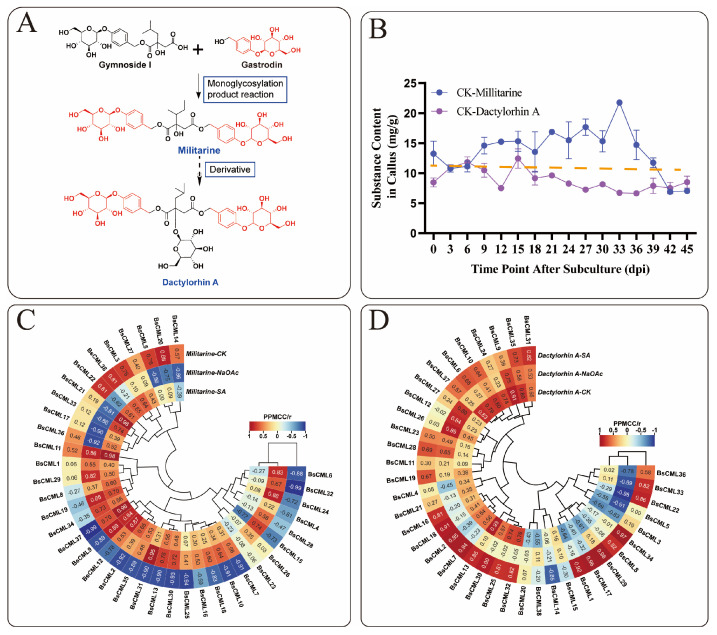
Correlation of *BsCML* expression with accumulations of militarine and dactylorhin A. (**A**) Relationship between militarine, gastrodin, and dactylorhin A. (**B**) The accumulation trend of militarine and the derivative dactylorhin A in the 0–45 dpi control group, with the dotted line representing the virtual symmetry axis. (**C**) Correlation heatmap of *BsCML* expression in *B. striata* callus and militarine synthesis. (**D**) Correlation Heatmap of *BsCML* expression in B.striata callus and dactylorhin A synthesis.

**Figure 9 plants-14-01052-f009:**
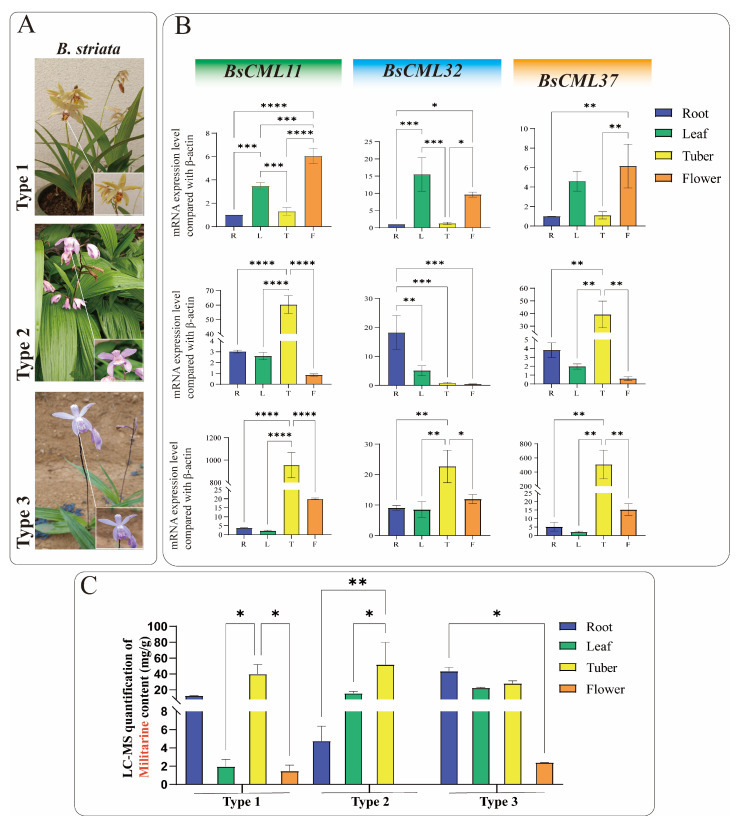
Expression levels of three *BsCML* genes and militarine content in three *B. striata* varieties. (**A**) Three *B. striata* varieties with distinct agronomic traits. (**B**) Expression profiles of three *BsCML* genes in different tissues of the three *B. striata* varieties. (**C**) LC-MS quantification of militarine in different tissues of the three *B. striata* varieties. Asterisks denote significance levels: * *p* < 0.05, ** *p* < 0.01, *** *p* < 0.001, **** *p* < 0.0001 (one-way ANOVA).

**Table 1 plants-14-01052-t001:** Physicochemical properties and cellular localization of BsCML family proteins.

Initial Sample ID	Gene Name	AA	MW/Da	pI	Instability Index	Aliphatic Index	GRAVY	Cell Localization
TRINITY_DN19138_c0_g1	*BsCML1*	179	20,577.79	5.24	35.34	94.80	−0.101	Outside the cell
TRINITY_DN26172_c0_g1	*BsCML2*	181	20,495.56	4.76	49.97	96.96	−0.101	Chloroplast
TRINITY_DN26571_c0_g1	*BsCML3*	110	12,768.39	4.42	71.19	68.27	−0.498	Nucleus
TRINITY_DN28075_c0_g1	*BsCML4*	200	22,455.40	4.79	44.41	77.10	−0.468	Chloroplast
TRINITY_DN2950_c0_g1	*BsCML5*	104	12,116.79	4.25	47.11	80.67	−0.348	Nucleoplasm
TRINITY_DN3140_c0_g1	*BsCML6*	190	20,739.08	4.50	41.73	70.37	−0.459	Nucleus
TRINITY_DN32186_c0_g1	*BsCML7*	143	16,315.59	4.35	44.51	79.79	−0.245	Cytoplasm
TRINITY_DN32287_c0_g1	*BsCML8*	71	7983.08	4.81	29.81	78.17	−0.142	Chloroplast
TRINITY_DN32287_c0_g2	*BsCML9*	197	21,681.42	4.59	54.57	87.61	−0.185	Chloroplast
TRINITY_DN32415_c0_g1	*BsCML10*	210	22,981.24	4.54	28.44	87.33	−0.050	Chloroplast
TRINITY_DN34106_c2_g1	*BsCML11*	169	19,061.87	4.60	72.05	84.26	−0.321	Chloroplast
TRINITY_DN34539_c0_g1	*BsCML12*	78	8942.06	4.36	55.61	87.56	−0.404	Chloroplast
TRINITY_DN34539_c0_g2	*BsCML13*	144	16,522.78	4.56	57.28	80.56	−0.351	Cytoplasm
TRINITY_DN34644_c0_g1	*BsCML14*	76	8570.50	4.29	60.26	53.95	−0.659	Nucleus
TRINITY_DN36231_c0_g3	*BsCML15*	151	17,416.62	4.24	31.87	83.77	−0.430	Chloroplast
TRINITY_DN38066_c5_g3	*BsCML16*	154	17,277.25	4.38	38.65	75.32	−0.584	Cytoplasm
TRINITY_DN40985_c3_g1	*BsCML17*	271	29,322.21	6.31	56.17	65.65	−0.368	Nucleus
TRINITY_DN44252_c6_g1	*BsCML18*	165	18,004.76	4.40	37.28	68.55	−0.594	Chloroplast
TRINITY_DN45768_c0_g1	*BsCML19*	175	20,078.39	4.76	57.66	64.57	−0.687	Mitochondria
TRINITY_DN46042_c1_g1	*BsCML20*	159	16,512.49	3.71	43.35	92.01	0.217	Chloroplast
TRINITY_DN48433_c0_g3	*BsCML21*	166	18,097.09	4.36	35.78	86.51	−0.308	Nuclear plastids
TRINITY_DN50047_c2_g2	*BsCML22*	171	19,347.59	4.70	19.69	70.82	−0.696	Nucleus
TRINITY_DN50169_c2_g7	*BsCML23*	157	17,540.53	4.42	32.96	89.55	−0.362	Nuclear plastids
TRINITY_DN50958_c2_g1	*BsCML24*	188	20,703.16	4.66	49.73	80.43	−0.354	Mitochondria
TRINITY_DN51047_c0_g1	*BsCML25*	176	19,973.81	5.00	52.25	80.91	−0.368	Chloroplast
TRINITY_DN51047_c0_g3	*BsCML26*	176	19,751.21	4.62	58.48	70.91	−0.470	Nucleus
TRINITY_DN51580_c0_g3	*BsCML27*	147	16,633.93	4.75	36.21	86.94	−0.329	Cytoplasm
TRINITY_DN51935_c2_g1	*BsCML28*	121	13,850.40	4.10	44.61	75.04	−0.440	Cytoplasm
TRINITY_DN54559_c2_g2	*BsCML29*	225	25,603.04	4.88	49.68	68.04	−0.457	Chloroplast
TRINITY_DN5534_c0_g1	*BsCML30*	209	22,598.21	5.05	59.48	64.40	−0.501	Chloroplast
TRINITY_DN55906_c1_g2	*BsCML31*	205	22,145.21	4.55	32.41	99.46	0.023	Chloroplast
TRINITY_DN58835_c0_g2	*BsCML32*	161	17,438.72	4.58	31.36	88.63	−0.012	Nucleus
TRINITY_DN5996_c0_g1	*BsCML33*	189	21,237.32	4.48	31.80	94.92	0.014	Chloroplast
TRINITY_DN60921_c0_g1	*BsCML34*	177	20,115.05	4.34	26.27	88.64	−0.006	Nucleus
TRINITY_DN60932_c0_g1	*BsCML35*	174	19,864.72	4.64	58.97	87.41	−0.235	Chloroplast
TRINITY_DN60945_c0_g1	*BsCML36*	113	13,140.75	4.14	54.13	71.50	−0.565	Cytoplasm
TRINITY_DN7103_c0_g1	*BsCML37*	178	20,429.54	4.74	38.91	77.19	−0.193	Chloroplast
TRINITY_DN7893_c0_g1	*BsCML38*	185	19,965.29	4.57	44.75	78.11	−0.328	Nucleus

**Table 2 plants-14-01052-t002:** EST-SSR Fragment Information.

Gene Name	Repeat	Forward Primer	Reverse Primer
*BsCML9*	(gga)5	GGCCTCGATTTCGACTCCTT	AATCCATCGCCGTCCTCATC
*BsCML11*	(ccg)4-(cct)4	CCGCCCTTCTACTCACCAAA	GTTGGCGTCTATGTGGTGGA
*BsCML15*	(ag)10	CCGTCTTGACCGGCTTTTAG	ACCACCAAGCCCACCATTTT
*BsCML17*	(tcc)3-(cag)3	AAAGACACAAGGAGGAGGCG	TAAGAAGAGAGAGCGCGCTG
*BsCML20*	(ccg)5	GAGCTGAAGGCCATCATCGA	CACTCATCATGCCATCCCCA
*BsCML23*	(ag)9	GATGACAGATCACCGGAGCC	AAGATCCCTCGGAGCTCAGT
*BsCML31*	(ct)11	—	—

Note: ‘—’ indicates no primer designed. Primers for *BsCML31* were not designed due to technical constraints (e.g., repetitive sequences).

**Table 3 plants-14-01052-t003:** RT-qPCR primer information for the *B. striata BsCML* gene family.

Name	Primer	Tm/C	GC%
*BsCML*_*4F*	TCCCACAGCTATCATCCCCA	60.0	55.0
*BsCML*_*4R*	AGCTCGGTCGGAGAGATCTT	60.1	55.0
*BsCML*_*6F*	GCAAATTCCGGTCGCTCTTC	59.9	55.0
*BsCML*_*6R*	GACGCGTTCAAGCTCGTTTT	60.0	50.0
*BsCML*_*16F*	CGCTTGGTTTGAAACAGGGG	60.0	55.0
*BsCML*_*16R*	CAAGTCAAGGCCGCAAATCC	60.1	55.0
*BsCML*_*27F*	GCGACGCTTTCAAAGTCCTC	59.8	55.0
*BsCML*_*27R*	GCGGATCCACTCATCGAACT	59.9	55.0
*β-actin*_F	AATCCCAAGGCAAACAGA	51.00	18.00
*β-actin*_R	CACCATCACCAGAATCCAG	53.00	19.00

Note: Primer information for *BsCML11* and *BsCML23* is provided in Table 2.

## Data Availability

The data presented in the study are deposited in the NCBI repository, accession number PRJNA1009214.

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
