# Peer review of "Transcriptome Analysis of the *CML* Gene Family in *Bletilla striata* and Regulation of Militarine Synthesis Under Sodium Acetate and Salicylic Acid Treatments"

_plants, 2025, doi:10.3390/plants14071052_

Round 1

Reviewer 1 Report

Comments and Suggestions for Authors

Dear authors,

  1. Lack of data: There is no support data presented for the result findings in subsection “2.1. Functional enrichment of BsCML”.
  2. Revise Supplementary Tables S1-S3 to Table 1-3 since these tables were directly presented in the main text of the manuscript, not in the separate files of Supplementary Materials.
  3. This statement should be revised (Line 224-227): “The results showed that  the expression patterns of  five differentially expressed genes (DEGs) were largely consistent with the transcriptome sequencing trend, thereby confirming the reliability of the transcriptome data.” For example, BsCML27, its expression pattern did not show a consistent trend between FPKM and qPCR. For FPKM expression patterns (Fig.8A),  its trend is in the following order: SA > control  > NaAc at 18 and 21 dpi. However, based on qPCR, its expression showed the highest in the control group, and the trend was the following order: control  > SA > NaAc.
  4. Lack of sequence of β-actin primer. Please provide the primer sequence of the β-actin reference gene in Table S3.
  5. Pay special attention to the genes, and proteins mentioned in the study, using italics to differentiate genes from proteins. For gene, it is present in italics but does not apply to protein. For instance, “CML” (Line 102), BsCML (L105), and many others in subsection 2.2 and elsewhere, denote protein. It should not be presented in italics. Please double-check the whole manuscript and revise it.
  6. Unclear of cited ref number #21 in the sentence (Line 460-462): “Total RNA was extracted from each sample [21] for transcriptome sequencing, and the CML gene set of B. striata was screened from the sequencing results for subsequent analysis.”

Other Remarks

- Represent the manuscripts following journal format: “Capitalize Each Word” in the title and subsections.

+ Title: Transcriptome analysis of the CML gene family in Bletilla striata and regulation of militarine synthesis under sodium acetate and salicylic acid treatments

-> Transcriptome Analysis of the CML Gene Family in Bletilla striata and Regulation of Militarine Synthesis under Sodium Acetate and Salicylic Acid Treatments

+ Subsection:

For example, “2.2. Identification and proteins physicochemical property analysis of BsCML

-> “2.2. Identification and Proteins Physicochemical Property Analysis of BsCML”

- Section “Reference” should be presented according  to  the  journal’s style and  include an abbreviated journal name, year, volume, and page range as journal format:

“Author 1, A.B.; Author 2, C.D. Title of the article. Abbreviated Journal Name Year, Volume, page range.”

For example, Ref #1:

 “1.  Ranty, B.;  Aldon, D.;  Cotelle, V.;  Galaud, J. P.;  Thuleau, P.; Mazars, C., Calcium Sensors as Key Hubs in Plant Responses to Biotic and Abiotic Stresses. Frontiers in plant science 2016, 7, 327. http://dx.doi.org/10.3389/fpls.2016.00327”

-> “1.  Ranty, B.;  Aldon, D.;  Cotelle, V.;  Galaud, J. P.;  Thuleau, P.; Mazars, C. Calcium sensors as key hubs in plant responses to biotic and abiotic stresses. Front. Plant Sci. 2016, 7, 327. http://dx.doi.org/10.3389/fpls.2016.00327”

Best regards,

Author Response

Dear Reviewer,

We sincerely appreciate your thorough review and valuable suggestions. Your comments are scientifically rigorous and logically sound, providing critical guidance for improving the quality of our manuscript. We have carefully addressed each point and made corresponding revisions to enhance the paper. We hope the revised version meets your approval and look forward to your further feedback. Below, we provide detailed responses to your specific comments:

Reviewer 1:

  1. Lack of data: There is no support data presented for the result findings in subsection “1. Functional enrichment of BsCML”.

Reply: We sincerely appreciate your feedback regarding the lack of supporting data in the section "2.1. Functional Enrichment of BsCML." Your comment has encouraged us to thoroughly reassess the completeness of this section. To clarify, the findings presented in section 2.1 are based on differential gene and metabolite data obtained from transcriptomic analysis, with functional annotation performed using the GO and KEGG databases. These analyses culminated in the functional description of BsCML genes, as detailed in lines 104-106 of the manuscript.

We recognize that the original presentation of these results relied predominantly on textual descriptions, which may have compromised the clarity of the data. To address this, we have updated the title of section 2.1 from "Functional Enrichment of BsCML" to "Functional Annotation of BsCML," highlighting the distinction between "Enrichment" and "Annotation" to better align with the content we have presented. Furthermore, we have refined the key terms in this section to ensure a more precise and scientifically robust description of our research. For these updates, please refer to lines 101 and 104 of the revised manuscript.

Additionally, the transcriptomic data and related supporting data have been uploaded to the NCBI database under the accession number PRJNA1009214, as noted in lines 838-839 of the manuscript. We believe these revisions and clarifications address your concerns and contribute to improving the readability and scientific quality of the paper.

2.Revise Supplementary Tables S1-S3 to Table 1-3 since these tables were directly presented in the main text of the manuscript, not in the separate files of Supplementary Materials.

Reply: Thank you for your suggestions, which have helped us clarify the nature and positioning of "Supplementary Tables S1-S3." Based on your feedback, we have made adjustments throughout the manuscript to address the related issues. Please refer to lines 116, 125, 140, 391, 590, 593, 729, and 733 of the revised manuscript for these updates.

3.This statement should be revised (Line 224-227): “The results showed that the expression patterns of five differentially expressed genes (DEGs) were largely consistent with the transcriptome sequencing trend, thereby confirming the reliability of the transcriptome data.” For example, BsCML27, its expression pattern did not show a consistent trend between FPKM and qPCR. For FPKM expression patterns (Fig.8A), its trend is in the following order: SA > control > NaAc at 18 and 21 dpi. However, based on qPCR, its expression showed the highest in the control group, and the trend was the following order: control > SA > NaAc.

Reply: We sincerely appreciate your constructive feedback. Upon careful consideration, we agree that the original conclusion in the manuscript was not sufficiently detailed and lacked some scientific rigor, particularly in addressing the discrepancy in expression trends of the BsCML27 gene between FPKM and qPCR data. Based on the results presented in Figure 6B, we have thoroughly re-examined this issue and revised the text in accordance with your suggestions to make it more scientifically sound.

Specifically, we have revised the statement from “The results showed that the expression patterns of five differentially expressed genes (DEGs) were largely consistent with the transcriptome sequencing trend, thereby confirming the reliability of the transcriptome data.” to: “The results showed that, among the six randomly selected genes, the expression trends of five genes were largely consistent between FPKM and qPCR data. However, an inconsistency was observed in the expression trend of BsCML27. For example, at 18 days, the FPKM expression pattern of BsCML27 was SA > control > NaAc, whereas the qPCR pattern was control > SA > NaAc. Overall, the qPCR validation results confirm the reliability of the transcriptomic FPKM data.” Please refer to line 274-280 of the revised manuscript for these updates.

4.Lack of sequence of β-actin primer. Please provide the primer sequence of the β-actin reference gene in Table S3.

Reply:We sincerely appreciate your reminder. We have carefully reviewed the entire manuscript and have supplemented the primer information for β-actin in Table 3. Please refer to Table 3 in line 759 of the manuscript for the updated details.

5.Pay special attention to the genes, and proteins mentioned in the study, using italics to differentiate genes from proteins. For gene, it is present in italics but does not apply to protein. For instance, “CML” (Line 102), BsCML (L105), and many others in subsection 2.2 and elsewhere, denote protein. It should not be presented in italics. Please double-check the whole manuscript and revise it.

Reply: Thank you very much for your meticulous and rigorous review. I have carefully re-examined the entire manuscript for the italicization of gene and protein names and made the necessary revisions as requested. Please refer to the following lines in the revised manuscript for the updates: 55-57, 104, 111, 113, 116, 118-131, 148, 152-161, 169-195, 231-244, 252-254, 416, 465-468, 471-473, 484, 573, 577-578, 583, 642, 705, 778, and 782.

  1. Unclear of cited ref number #21 in the sentence (Line 460-462): “Total RNA was extracted from each sample [21] for transcriptome sequencing, and the CML gene set of B. striata was screened from the sequencing results for subsequent analysis.”

Reply: Thank you for raising this question. We would like to clarify that reference [21] serves as the experimental foundation for this project, and its citation here is intended to indicate that we referenced its experimental methodology. To avoid ambiguity in the citation, we have moved the reference "[21]" to the preceding sentence, which now reads: " After the second subculture, 150 μmol/L NaAc and 15 μmol/L SA were added separately, and three samples were randomly selected at 3 dpi, 18 dpi, 21 dpi, and 36 dpi [21]." Please refer to line 554 of the revised manuscript for this adjustment.

  1. Represent the manuscripts following journal format: “Capitalize Each Word” in the title and subsections.

+ Title: Transcriptome analysis of the CML gene family in Bletilla striata and regulation of militarine synthesis under sodium acetate and salicylic acid treatments

-> Transcriptome Analysis of the CML Gene Family in Bletilla striata and Regulation of Militarine Synthesis under Sodium Acetate and Salicylic Acid Treatments

+ Subsection: For example, “2.2. Identification and proteins physicochemical property analysis of BsCML”

-> “2.2. Identification and Proteins Physicochemical Property Analysis of BsCML”

Reply: Based on your comments regarding the capitalization of words in the title and the journal's requirements, we have carefully revised it. Please refer to the updated version in lines 2-4.

In addition, regarding the issue of capitalization in the title, we have also carefully revised the capitalization of all headings throughout the manuscript. Please refer to the revised lines 101, 111, 139, 156, 194, 209, 239-240, 291, 333, 384, 404, 458, 507, 526, 540-541, 563, 570, 578, 598-600, 609, 619, 626, 627, 732, and 744 in the revised manuscript for details.

  1. Section “Reference” should be presented according to the journal’s style and include an abbreviated journal name, year, volume, and page range as journal format: “Author 1, A.B.; Author 2, C.D. Title of the article. Abbreviated Journal Name Year, Volume, page range.”

For example, Ref #1: “1.  Ranty, B.; Aldon, D.; Cotelle, V.; Galaud, J. P.; Thuleau, P.; Mazars, C. Calcium Sensors as Key Hubs in Plant Responses to Biotic and Abiotic Stresses. Frontiers in plant science 2016, 7, 327. http://dx.doi.org/10.3389/fpls.2016.00327”

-> “1.  Ranty, B.; Aldon, D.; Cotelle, V.; Galaud, J. P.; Thuleau, P.; Mazars, C. Calcium sensors as key hubs in plant responses to biotic and abiotic stresses. Front. Plant Sci. 2016, 7, 327. http://dx.doi.org/10.3389/fpls.2016.00327”

Reply: Thank you very much for pointing out the issues with the reference formatting. Following the journal's guidelines and your suggestions, I have revised the formatting of all 63 references, including adjustments to journal name abbreviations and title capitalization. Please refer to lines 844-1007 of the revised manuscript for these updates.

We would like to express our sincere gratitude to you and the reviewers for your valuable time, insightful comments, and constructive feedback on our manuscript titled “Transcriptome Analysis of the CML Gene Family in Bletilla striata and Regulation of Militarine Synthesis under Sodium Acetate and Salicylic Acid Treatments.” We have carefully considered all of your kind suggestions and have revised the manuscript accordingly to enhance its quality and comprehensiveness. We believe that the revisions have substantially improved the manuscript, addressing all the concerns raised by the reviewers and enhancing the overall quality of our research. We are confident that the revised manuscript now meets the standards for publication in Plants.

Should you require any further information or additional revisions, please do not hesitate to contact us.

Sincerely yours,

Prof. Dr. Delin Xu

March 24, 2025

Reviewer 2 Report

Comments and Suggestions for Authors

The manuscript by Li et al. presents a comprehensive analysis of the BsCML gene family in Bletilla striata, offering functional insights into militarine biosynthesis and elicitor responses, highlighting their potential role in plant stress adaptation. By integrating bioinformatics, gene expression analysis, and metabolite quantification, the study adopts a multi-dimensional approach to understanding BsCML gene functions. Linking BsCML genes to metabolite biosynthesis lays a foundation for future genetic engineering and breeding efforts to enhance medicinal compound production in B. striata.

While the study is thorough, several limitations should be addressed:

 The findings rely primarily on correlation-based evidence without direct functional validation through gene expression modifications. Generating plant lines with silenced, knocked-out, or overexpressed BsCML genes would provide stronger evidence for their role in secondary metabolite biosynthesis.

Additionally, while BsCML genes are compared to those in Arabidopsis, expanding the analysis to other metabolite-rich medicinal plants would strengthen the discussion. Current functional predictions are largely based on homology, lacking experimental validation.

Some additional suggestions:

The study also demonstrates that NaAc and SA influence metabolite accumulation but does not fully explore the underlying mechanisms. Further investigation into hormone signaling pathways and calcium-mediated stress responses would enhance the mechanistic understanding of these elicitor effects. A deeper co-expression analysis of BsCML genes with other metabolic pathway genes could provide additional insights into their regulatory roles. While this may not be feasible in the current study, future metabolomic profiling under varying stress conditions (e.g., drought or heavy metal exposure) could further elucidate BsCML-mediated regulation.

Finally, additional assays to determine whether NaAc and SA function through known calcium-dependent signaling pathways, along with a more detailed time-course analysis of BsCML expression, would clarify the dynamics of elicitor-induced gene regulation.

From a technical perspective, I did not identify any obvious typographical or formatting errors. The manuscript is well-written, with clear and concise language. The figures and figure captions are high-resolution and well-explained, effectively complementing the text.

Overall, the manuscript provides valuable insights into the role of BsCML genes in secondary metabolite biosynthesis in B. striata. Addressing the need for functional validation and deeper mechanistic analyses would significantly enhance its impact and broader relevance to medicinal plant research.

Author Response

Dear Reviewer,

We sincerely appreciate your thoughtful review and valuable suggestions regarding our manuscript. Your comments have provided important insights that will significantly improve both our current work and planned future research. We are particularly grateful for your recognition and support, which have greatly encouraged us and strengthened our confidence in conducting further studies. In response to your specific comments, we provide the following detailed clarifications:

Before addressing specific comments, we would like to contextualize our research background. Our team has accumulated nearly a decade of dedicated exploration in genetic breeding of Chinese medicinal herbs, with particular focus on Bletilla striata. Recent investigations have centered on militarine, a bioactive secondary metabolite demonstrating multifaceted pharmacological activities. Our ultimate objective is to enhance the therapeutic quality of B. striata through systematic metabolic engineering. Building upon this foundation, we conducted preliminary investigations combining integrated transcriptomic and metabolomic analyses. These efforts identified several gene families potentially involved in militarine biosynthesis pathways, particularly the BsCML family as referenced in our manuscript. As you astutely observed, “By integrating bioinformatics, gene expression analysis, and metabolite quantification, the study adopts a multi-dimensional approach to understanding BsCML gene functions", we have initiated functional characterization of BsCML proteins to assess their biotechnological potential and prioritize targets for subsequent research phases.

Reviewer 2:

  1. The findings rely primarily on correlation-based evidence without direct functional validation through gene expression modifications. Generating plant lines with silenced, knocked-out, or overexpressed BsCML genes would provide stronger evidence for their role in secondary metabolite biosynthesis.

Reply: We sincerely appreciate your insightful comments regarding the importance of functional validation in our study. We fully agree that direct functional validation through gene expression modifications, such as silencing, knockout, or overexpression of BsCML genes, would significantly strengthen the evidence supporting their roles in secondary metabolite biosynthesis.

While in our present study, we conducted a comprehensive transcriptome analysis to identify and characterize the BsCML gene family in Bletilla striata. Through expression profiling under sodium acetate (NaAc) and salicylic acid (SA) treatments, we established strong correlative evidence linking specific BsCML genes (notably BsCML32 and BsCML37) to militarine synthesis. These findings provide a foundational understanding of the potential regulatory roles of BsCML genes in secondary metabolite pathways.

At this stage, due to the specific constraints of time constraints, we have not yet been able to perform the proposed functional validation experiments. Particularly, B. striata presents certain challenges for genetic manipulation, which we are actively addressing to facilitate future studies.

We recognize the importance of functional validation and have planned subsequent experiments to generate transgenic B. striata lines with targeted modifications of BsCML genes. These studies will aim to directly assess the impact of BsCML gene alterations on militarine production and overall plant physiology. We anticipate that these future investigations will provide the mechanistic insights necessary to confirm the roles of BsCML genes suggested by our current correlation-based findings.

Despite the absence of direct functional validation at this time, our study offers robust correlative data supported by evolutionary analyses and expression profiling. The significant correlations (P<0.01) between BsCML gene expression and metabolite accumulation under specific treatments provide compelling evidence for their involvement in secondary metabolite biosynthesis. We believe that these results contribute meaningful insights to the field and lay the groundwork for subsequent functional studies.

We do appreciate your kind suggestion and acknowledge that functional validation is a crucial next step. We are committed to pursuing these experiments and are confident that future work will substantiate and expand upon the findings presented in our current manuscript. Thank you for your thoughtful consideration and constructive feedback.

  1. Additionally, while BsCML genes are compared to those in Arabidopsis, expanding the analysis to other metabolite-rich medicinal plants would strengthen the discussion. Current functional predictions are largely based on homology, lacking experimental validation.

Reply: Thanks a lot for your insightful and constructive comment. We fully acknowledge that expanding our comparative analysis to include BsCML genes from other metabolite-rich medicinal plants would provide a more comprehensive understanding of their evolutionary conservation and functional diversification in secondary metabolism. In our manuscript, we focused on comparing BsCML genes primarily with those in Arabidopsis thaliana due to the extensive genetic and functional annotation available for this model organism. Arabidopsis serves as a foundational reference for plant gene family studies, allowing us to draw initial functional predictions based on well-characterized homologs.

Besides, in our current study, we met several limitations which blocking us to do a more comprehensive analysis. The first is that at the time of our study, comprehensive and high-quality genomic and transcriptomic data for other metabolite-rich medicinal plants were limited. This constraint hindered our ability to perform an extensive comparative analysis beyond Arabidopsis. Secondly, our primary aim was to characterize the BsCML gene family in Bletilla striata and investigate their potential roles in militarine synthesis. Expanding the comparative analysis was considered, but we opted to maintain a focused approach to ensure depth and clarity in our findings.

While respecting your valuable suggestions, we plan to do more analysis in our upcoming studies. It includes additional comparative analyses with other metabolite-rich medicinal plants such as Panax ginseng, Withania somnifera, and Camellia sinensis, provided that sufficient genomic data becomes available. And with the continuous advancement and publication of genomic data for various medicinal plants, we will update our comparative analyses to encompass a broader range of species, thereby strengthening our functional predictions through enhanced homology-based correlations.

We greatly appreciate the reviewer’s recommendation to expand our comparative analysis to other medicinal plants. This will undoubtedly enhance the breadth and impact of our research. We are committed to integrating this valuable suggestion into our future work and believe that it will significantly contribute to the understanding of CML gene functions in medicinal plant secondary metabolism.

  1. The study also demonstrates that NaAc and SA influence metabolite accumulation but does not fully explore the underlying mechanisms. Further investigation into hormone signaling pathways and calcium-mediated stress responses would enhance the mechanistic understanding of these elicitor effects.

Reply: We sincerely appreciate your comments on highlighting this important aspect of our study. We acknowledge that while our research effectively demonstrates the influence of sodium acetate (NaAc) and salicylic acid (SA) on metabolite accumulation in B. striata, we focused on identifying and characterizing the BsCML gene family and analyzing their expression profiles under NaAc and SA treatments. Our findings indicated that BsCML32 and BsCML37 are closely associated with militarine synthesis, suggesting a role in the regulation of secondary metabolite biosynthesis in response to these elicitors. This correlation provides valuable insights into the potential signaling pathways involved.

With your kind reminder, we have realized the limitations of our current study, trapping in time constraints and technical challenges. Therefore, we wholeheartedly agree with your comments on the necessity of investigating the underlying mechanisms to fully understand how NaAc and SA regulate metabolite accumulation. Addressing these mechanisms is a priority in our ongoing and future research endeavors. We are confident that exploring these pathways will significantly advance our understanding of the regulatory networks governing secondary metabolite biosynthesis in B. striata.

  1. A deeper co-expression analysis of BsCML genes with other metabolic pathway genes could provide additional insights into their regulatory roles. While this may not be feasible in the current study, future metabolomic profiling under varying stress conditions (e.g., drought or heavy metal exposure) could further elucidate BsCML-mediated regulation.

Reply: I have carefully reviewed your comments and found them to be highly valuable for guiding the future direction of this research. As you rightly pointed out, an in-depth analysis of the regulatory role of BsCML through co-expression and metabolomics studies under diverse stress conditions would significantly enhance the depth and impact of this work. I greatly appreciate your insightful suggestion.

During our research, we observed that the CML gene family plays a broad and critical role in plant stress responses, which suggests that exploring its functions under varied stress conditions could yield many intriguing findings. We plan to build upon the foundation established in this manuscript and further investigate these aspects in our future work. Incorporating these approaches will undoubtedly enrich our understanding of BsCML-mediated regulation in B. striata and advance the field of plant secondary metabolism. We are committed to integrating your valuable suggestions into our future research endeavors, thereby enhancing the depth and impact of our studies.

  1. Finally, additional assays to determine whether NaAc and SA function through known calcium-dependent signaling pathways, along with a more detailed time-course analysis of BsCML expression, would clarify the dynamics of elicitor-induced gene regulation.

Reply: Thank you for your valuable comments. We would like to clarify that in Line 316 of the manuscript (Figure 7), we demonstrated the effects of NaAc and SA treatment on B. striata callus, showing that these elicitors significantly influence its growth and metabolic processes. Additionally, in Line 281 (Figure 6A), we presented the expression patterns of BsCML under control, NaAc, and SA treatments. Our findings reveal that these elicitors either inhibit or promote the expression of specific BsCML genes. For instance, compared to the control group, SA treatment induced the expression of BsCML27 at 18 dpi, while NaAc treatment suppressed the expression of BsCML18 at 3 dpi. These results suggest that NaAc and SA are involved in modulating the calcium signaling pathway. However, we fully acknowledge that direct mechanistic evidence is still lacking, and we plan to address this gap in our future studies.

Besides, in this manuscript, we selected 3 dpi, 18 dpi, 21 dpi, and 36 dpi as time points for investigation based on previous research. However, we recognize that these time points are insufficient for a detailed time-series analysis of BsCML. To address this limitation, we plan to supplement this work in future studies by including earlier time points, such as 3 h, 6 h, and 12 h, under drought or heavy metal stress conditions for a more comprehensive analysis.

We sincerely thank you once again for your support and recognition of this work, as well as for your valuable suggestions, which have greatly inspired our thinking about future research directions. Your insights have been immensely beneficial to us. Building on this foundation, we plan to enrich the functional verification and mechanistic analysis in a new research topic, thereby enhancing the universality and impact of this study.

Once again, we do thank you for the opportunity to revise our work. We appreciate the time and effort invested by you and the reviewers to help us improve our manuscript. We look forward to your favorable consideration.

Sincerely yours,

Prof. Dr. Delin Xu

March 24, 2025